# Apoptotic Activity of New Oxisterigmatocystin Derivatives from the Marine-Derived Fungus *Aspergillus nomius* NC06

**DOI:** 10.3390/md19110631

**Published:** 2021-11-11

**Authors:** Muh. Ade Artasasta, Yanwirasti Yanwirasti, Muhammad Taher, Akmal Djamaan, Ni Putu Ariantari, Ru Angelie Edrada-Ebel, Dian Handayani

**Affiliations:** 1Laboratory of Sumatran Biota, Faculty of Pharmacy, Andalas University, Padang 25163, Indonesia; muh.ade.artasasta.fmipa@um.ac.id (M.A.A.); akmaldjamaan@phar.unand.ac.id (A.D.); 2Biotechnology Department, Faculty of Mathematics and Natural Sciences, Universitas Negeri Malang (UM), Malang 65145, Indonesia; 3Departement of Biomedical, Faculty of Medicine, Andalas University, Padang 25163, Indonesia; yanwirasti@yahoo.com; 4Faculty of Pharmacy, International Islamic University Malaysia, Kuantan 25200, Malaysia; mtaher@iium.edu.my; 5Department of Pharmacy, Faculty of Mathematics and Natural Sciences, Udayana University, Bali 80361, Indonesia; putu_ariantari@unud.ac.id; 6Strathclyde Institute of Pharmacy and Biomedical Sciences, University of Strathclyde, The John Arbuthnott Building, 161 Cathedral Street, Glasgow G4 0RE, UK; ruangelie.edrada-ebel@strath.ac.uk

**Keywords:** marine sponge-derived fungus, *Neopetrosia chaliniformis*, *Aspergillus nomius*, oxisterigmatocystin, cytotoxic activity, HT 29 colon cancer cells, apoptosis cells

## Abstract

Sponge-derived fungi have recently attracted attention as an important source of interesting bioactive compounds. *Aspergillus nomius* NC06 was isolated from the marine sponge *Neopetrosia chaliniformis*. This fungus was cultured on rice medium and yielded four compounds including three new oxisterigmatocystins, namely, J, K, and L (**1**, **2**, and **3**), and one known compound, aspergillicin A (**4**). Structures of the compounds were elucidated by 1D and 2D NMR spectroscopy and by high-resolution mass spectrometry. The isolated compounds were tested for cytotoxic activity against HT 29 colon cancer cells, where compounds **1**, **2**, and **4** exhibited IC_50_ values of 6.28, 15.14, and 1.63 µM, respectively. Under the fluorescence microscope by using a double staining method, HT 29 cells were observed to be viable, apoptotic, and necrotic after treatment with the cytotoxic compounds **1**, **2**, and **4**. The result shows that compounds **1** and **2** were able to induce apoptosis and cell death in HT 29 cells.

## 1. Introduction

In recent years, sponge-derived fungi have drawn increasing attention as a rich source of interesting bioactive marine natural products [1,2,3,4,5,6]. Many fungi associated with marine sponges produce interesting secondary metabolites such as alkaloids, polyketides, and terpenoids [6,7,8,9]. This has encouraged many researchers to explore more bioactive compounds from marine sponge-derived fungi, potentially used as antimicrobials or as anticancer drugs and against other diseases [9,10,11,12,13]. The fungus *Aspergillus nomius* NC06 was isolated from the marine sponge *Neopetrosia chaliniformis*. The fungal ethyl acetate extract showed cytotoxic activity against WiDr and HCT 116 colon cancer cells. However, it was not toxic against the normal Vero cell line [14,15].

Extracts and secondary metabolites yielded by the fungal genus *Aspergillus* have been frequently reported in the literature for their potent cytotoxic activity [16,17]. One family of compounds described from this genus is the oxisterigmatocystins [18]. Oxisterigmatocystins A, B, and C, as well as 5-methoxysterigmatocystin, were isolated from *A. versicolor.* 5-Methoxysterigmatocystin exhibited moderate cytotoxicity against the A-549 and HL-60 cell lines, with IC_50_s of 3.86 and 5.32 µM, respectively [18]. Oxisterigmatocystin D was also reported from *Aspergillus* sp. with an additional methoxyl group in C-4′ (*δ*_H_ 3.10; *δ*_C_ 54.5) [19]. Other derivatives (E, F, G, H) were isolated from the fungus *Botryotrichum piluliferum* [20], and oxisterigmatocystin I was isolated from the fungus *Aspergillus* sp. F40 derived from the sponge *Callyspongia* [21]. In our continuing study, new oxisterigmatocystin derivatives J, K, and L (**1**, **2**, and **3**) and one known compound, aspergillicin A (**4**) [22], were isolated from the ethyl acetate extract of *A. nomius* NC06.

## 2. Results and Discussion

Compound **1** was isolated as a yellow amorphous powder. Its molecular formula was determined as C_19_H_14_O_7_ based on an HR-ESI-MS ion peak at *m*/*z* 355.0811 [M + H]^+^ calculated at 355.0812 Da for C_19_H_15_O_7_. The 1D NMR data of this compound (Table 1) contained resonances for one carbonyl group, nine quaternary carbons, five methine units (CH), two methoxys (O-CH_3_), and one hydroxyl group (Figure 1). Analysis of 1D and 2D NMR showed that compound **1** was an oxisterigmatocystin derivative [18]. The isolated compound **1** had comparable ^1^H NMR spectral data (Appendix A) to those of oxisterigmatocystin C [18] on the aromatic region with a singlet for H-2 at 6.62 ppm and a similar ABC system at *δ*_H_ 6.94, dd (8.5, 0.9); 7.65, t (8.4), and 7.08, dd (8.4, 0.9) for H-5, H-6, and H-7, respectively. In comparison to oxisterigmatocystin C, there was the loss of an exchangeable resonance at 13.4 ppm due to the methylation of the hydroxyl substituent on C-8 in compound **1**, which was replaced by a methoxy singlet at *δ*_H_ 3.85 with the corresponding ^13^C signal at 56.1 ppm. The major structural difference between compound **1** and the earlier described oxisterigmatocystin derivatives was the demethylation of the hydroxyl substituent at C-4′ in compound **1** and the additional olefinic methine signal for C-3′ at *δ*_H_ 6.77, d (*J* = 2.8 Hz) and *δ*_C_ 148.3 that was established by HSQC (Appendix A). Moreover, the existence of a double bond between the shielded quaternary C-2′ at *δ*_C_ 89.2 and C-3′ in compound **1** could be confirmed by the COSY and HMBC correlations with the hydroxylated methine unit on C-4′ at *δ*_H_ 5.71, d (*J* = 2.8 Hz) and *δ*_C_ 107.0, as well as with the C-4′O*H* singlet at 6.50 ppm, as shown in Figure 2 (Appendix A).

By modifying the Mosher method [23], the absolute stereochemistry on C-4′ was established to have the *R* configuration (Appendix A). Compound **1** was derivatized with chiral reagents *R*- and *S*-methoxy-α-(trifluoromethyl)phenylacetyl chloride (MTPA-Cl). The ^1^H-NMR spectrum for both congeners was measured, and the differences (Δ^δ(*S*)–δ(*R*)^) in chemical shifts for H-1′ and H-3′ in the respective spectra of the derivatized components were recorded. The ^1^H assignments for the MTPA esters were verified by COSY. H-1′ and H-3′ afforded a Δδ*^S-R^* of >0 and <0, respectively. The Mosher results apparently assigned the respective positions of H-1′ and H-3′ at the right and left sides of the *R*-hydroxyl stereocenter on C-4′, while the stereochemistry at position C-1′ remained undetermined. The observed nOe was inconclusive because in both cases, whether H-1′ has the *R* or *S* configuration, the through-space distances between H-1′ and H-4′ are 3.814 and 3.448Å, respectively, which are both less than the internuclear separation of 5Å that is the minimum requirement to identify spatially close pairs of nuclei. These through-space distances were determined by MM2 calculation at a minimum energy of 84.5646 and 81.4504 kcal/mol, respectively, using PerkinElmer’s 3D ChemDraw v 18.2. We assigned compound **1** the trivial name oxisterigmatocystin J.

Compounds **2** and **3** were also isolated as yellow amorphous powders. The molecular formulae for both compounds were established by HR-ESI-MS as C_20_H_18_O_7_ affording molecular ion peaks at [M + H]^+^, *m*/*z* 371.1127 and 371.1128, respectively, which were both calculated for C_20_H_19_O_7_ at 371.1125 Da. The aromatic regions of the ^1^H NMR spectral data of both **2** and **3** were comparable to those of the isolated compound **1** and oxisterigmatocystin C (Table 1). A methine singlet for H-2 was observed at 6.50 and 6.55 ppm for compounds **2** and **3**, respectively, while, again, a similar ABC system was observed for H-5, H-6, and H-7 in both congeners **2** and **3** at *δ*_H_ 7.02/7.04 (d, *J* = 8.4/8.3 Hz), 7.61/7.62 (t, *J* = 8.3/8.4 Hz), and 6.93 (d, *J* = 8.4/8.3 Hz), respectively, with differences in chemical shifts between ±0.01 and 0.02 ppm between the two isolated derivatives. As in compound **1**, C-1- and C-8-O*H* were methylated (Appendix A) as observed at 3.82 and 3.85 ppm, respectively. The ^13^C NMR spectral data of the xanthone moiety for the congeners **1**, **2**, and **3** were comparable, with an average chemical shift difference of 0.74 ppm, as shown in Table 1. This resulted in almost superimposable 2D NMR spectral data for HSQC and HMBC (Appendix A) for the compounds’ xanthone unit.

Structural differences between compounds **2** and **3** were evident at the 4′-methoxyhexahydrofuro [2,3-*b*]furan ring. The relative configurations of compounds **2** and **3** were resolved by comparison of their coupling constants, optical rotations, and conformational analysis with known congeners described in the literature [18]. The coupling constant between H-1′ and H-2′ of compounds **2** and **3** was at 6 Hz (with actual values at 6.1 and 5.9 Hz, respectively) that implied the *cis* configuration for the two vicinal protons and indicated a 1′*S*, 2′*S* stereochemistry [18].

Compound **2** afforded similar ^13^C and ^1^H NMR spectral data, which diverged at less than 0.1 and 0.5 ppm to those of oxisterigmatocystin C, respectively, while the coupling constants were also comparable at a maximum difference of 0.4 Hz between both derivatives, which indicated an identical spatial conformation. The β-configuration of the methoxy unit on C-4′ of compound **2** was similar to that found in oxisterigmatocystins A and C [18], which was based on the coupling pattern observed for an α-hemiacetal sterigmatocystin [18,24]. The coupling constant between H-2′ at 4.19 ppm and H-3′A at 2.39 ppm gave 9 Hz (with actual values at 9.3 and 9.5 Hz, respectively) for a dihedral angle of about 20° [24]. Albeit there was no observable coupling between H-3′B and H-2′, their dihedral angle must be between 80° and 100° as earlier described [24]. Furthermore, H-3′A displayed a geminal coupling of 13.3 Hz and a vicinal coupling of 5.0 Hz with H-4′ at 5.25 ppm for an expected dihedral angle of about 40°. The relative configurations for compound **2** were determined as 1′*S*, 2′*S,* and 4′*R* as in oxisterigmatocystin C [18,21]. We named compound **2** oxisterigmatocystin K, which is the 8-methoxy derivative of oxisterigmatocystin C [18].

On the other hand, the relative configuration of 4′-OMe in compound **3** was compatible with a β-hemiacetal sterigmatocystin [24] that is identical to oxisterigmatocystins B [18] and I [21], as evidenced by their comparable ^13^C and ^1^H NMR spectral data. The coupling constant between H-1′ at 6.48 and H-2′ at 4.24 ppm was 5.9 Hz. Moreover, H-2′ to H-3′A and H-3′B at 2.35 and 2.25 ppm coupled at 3.4 and 9.2 Hz, which would imply a dihedral angle of 20° and 120°, respectively, while H-3′A and H-3′B coupled with H-4′ at 5.17 ppm coupled at 5.0 Hz, which are not larger than 120° but both nearer to 60°. From the Noesy cross-peaks (Appendix A), correlations were observed between H-1′and H-2′, establishing the *cis* fusion for the bisfuran unit, as well as between H-2′ and 4-‘OMe. Therefore, as with oxisterigmatocystin B, the relative configurations for compound **3** were inferred as 1′*S*, 2′*S*, and 4′*S*. Consecutively, compound **3** was assigned the trivial name oxisterigmatocystin L, which is an 8-methoxy congener of oxisterigmatocystin I [21] that is a 5-demothoxylated derivative of oxisterigmatocystin B [18].

The structural diversity of the fungal metabolites oxisterigmatocystins has been predominantly based on the configuration of the hydroxyl or methoxy substituent on C-4′ [18,20,21,24]. When the C-4′ substituent has the β-configuration as in an α-hemiacetal sterigmatocystin, the bisfuran ring (C-1′ to C-4′) has an envelope conformation, and H-4′ is observed as a doublet. However, when the C-4′ substituent has the α-configuration for a β-hemiacetal sterigmatocystin, the bisfuran ring has a half-chair conformation, while H-4′ emerges as a triplet or a doublet of a doublet.

In this study, all four isolated compounds were tested for their cytotoxic activity (IC_50_) against HT 29 colon cancer cells (Table 2). Oxisterigmatocystins J and K along with the peptide aspergillicin A were found to be active, while oxisterigmatocystin L was inactive. The three oxisterigmatocystin congeners (J, K, and L) mainly differed on the hydroxyhexahydrofuro [2,3-*b*]furan moiety. The difference in the activity of the three sterigmatocystin derivatives was probably due to a change in spatial conformation on the hydrofuro [2,3-*b*]furan ring. There was a total loss of cytotoxicity in compound **3** with its β-configuration. Compounds **1** and **2** were bioactive, with IC_50_ values below 15 µg/mL. Alternatively, compound **2** had an α-configuration, which could be essential to the bioactivity of oxisterigmatocystins. However, as reported by Cai et al. 2011, the change in the spatial conformation of the methoxy substituent at C-4′ did not affect the cytotoxicity of the compounds as both α and β congeners were found to be inactive. Instead, as earlier proposed, the presence of a double bond on the methoxy furan ring (C-3′ and C-4′) was suspected to be the main cause of its cytotoxic activity against cancer cell lines A-549 and HL-60. In the case of oxisterigmatocystin J, in this study, the double bond on C-2′ and C-3′ could have been the basis of its cytotoxicity. It could also be further hypothesized that the cytotoxicity of oxisterigmatocystins must be specific to certain cancer cell lines.

All compounds did not show antibacterial activity against *E. coli* and *S. aureus*. This activity was influenced by bacterial resistance, susceptibility, persistence, and tolerance, the host factor, and the concentration of the compound [25].

Analysis of HT 29 cell death upon treatment with cytotoxic compounds **1**, **2**, and **4** was performed by a double staining method to determine the mechanism of cell death either by apoptosis or necrosis. The process of cell death in this test is expected to occur by apoptosis or programmed cell death due to the induction of the cytotoxic compounds. HT 29 cells were stained with acridine orange (AO) and propidium iodide (PI) to identify and quantify the viable cells versus cell death by apoptosis and necrosis and then observed under a fluorescence microscope after 24 h of exposure with the compounds. AO intercalates into DNA which gives a green fluorescence to the viable cells, while PI is only taken up by non-visible cells which intercalate into the DNA and give an orange fluorescence. Apoptotic cells have an orange to red nucleus with condensed or fragmented chromatin. Necrotic cells display a uniform red nucleus with a condensed structure [26,27,28].

As shown in Figure 3, an intact nucleus and membrane for untreated HT 29 cells were observed with a bright green fluorescence at 24 h. The fluorescence produced from HT 29 cells after the addition of AO–PI was generally of a uniform bright green color. The uniform bright green fluorescence in the nucleus is possessed by living cells that still have intact cell membranes. The treated HT 29 cells showed a non-uniform fluorescence color, a mixture of green with yellow fluorescence, which indicated apoptosis, and reddish-orange fluorescence for cells undergoing necrosis [26].

The apoptotic cells were quantified (as shown in Table 3) by calculating the percentage of viable cells, apoptotic cells, and necrotic cells from a total of 200 cells observed under the fluorescence microscope. The quantification of apoptotic cells is described as the percentage of apoptotic cells within the overall cell population. Compound **1** was able to induce the apoptosis of HT 29 with an apoptotic percentage of 30.65%, while the necrotic percentage was only 4.81%. In parallel, compound **2** exhibited an apoptotic and necrotic percentage of 59.38 and 4.12%. However, in contrast, compound **4** showed a higher percentage of necrotic cells at 69.85% than the percentage of apoptotic cells at 4.80%. The observed difference between the percentage of apoptotic and necrotic cells was significant at *p* < 0.05. The tests results carried out on HT 29 cells with oxisterigmatocystins J (**1**) and K (**2**), which both contained a xanthone nucleus and bisfuran structure, indicated the activation of apoptosis. On the other hand, aspergicillin A (**4**), which is a peptide, could not stimulate apoptosis. Based on these structural differences, it can be concluded that the xanthones and bisfuran nuclei were essential structural moieties capable of activating programmed cell death.

Some studies have reported that fungal cytotoxic compounds such as those explored by Yeh et al. (2009) from *Antrodia camphonata*, which included methyl antcinate, zhankuic acid A, and zhankuic acid C, promoted apoptosis of HT 29 cells at 41.7%, 32.7%, and 29.5%, respectively. Moreover, Schmelz et al. (1997) reported fumonisin B1 from *Fusarium moniliforme* that was able to activate the apoptosis of HT 29 cells at only 16% [29,30].

## 3. Conclusions

Chemical investigation of the EtOAc extract of the marine sponge-derived fungus *Aspergillus nomius* NC06 from Mandeh Island, West Sumatra, Indonesia, yielded four compounds, including three new oxisterigmatocystin congeners, namely, J (**1**), K (**2**), and L (**3**), along with one known compound, aspergillicin A (**4**). The bifuroxanthenone derivatives **1** and **2** with the peptide compound **4** were found to exhibit significant cytotoxicity against HT 29 colon cancer cell lines, with IC_50_ values of 6.28, 15.14, and 1.63 µM, respectively. The difference in the activity of the three sterigmatocystin derivatives was probably due to a change in spatial conformation and/or the presence of a double bond on the bisfuran ring. There was a total loss of cytotoxicity in compound **3** with its β-configuration. However, the xanthones and bisfuran nuclei in compounds **1** and **2** were thought to play a vital role in inducing cell apoptosis, as demonstrated through a double staining method. In summary, this study reveals the cytotoxic activity of secondary metabolites from the marine-derived fungus *A. nomius* NC06.

## 4. Materials and Methods

### 4.1. General Procedures

The isolation of compounds from *A. nomius* NC06 involved several instruments including chromatographic methods consisting of both analytical and semipreparative HPLC, NMR spectroscopy, and ESIMS. HPLC analysis was carried out on a Dionex Ultimate 3000 system with a C_18_ column, 4.6 × 150 mm, 5 µm, mobile phase A: H_2_O with 0.1% TFA; B: methanol, flow rate: 1 mL/min. Gradient elution from 0 to 35 min was 10–100% B, then washed at 100% B from 35 to 45 min and equilibrated back to 10% B from 46 to 60 min. Semipreparative HPLC was carried out with a Merck Hitachi Chromaster HPLC system. The chromatography method was conducted with silica gel 60 M for VLC and Sephadex LH-20 for column chromatography. The purity of the isolated compound was monitored by TLC using silica gel 60 F_254_ by using DCM and 5% MeOH as mobile phase. The 1D and 2D NMR spectra were recorded on a Bruker AVANCE DMX 600 NMR spectrometer. The chemical shifts (*δ*) were referenced to the residual solvent signals (DMSO-*d*_6_: *δ*_H_ 2.50/*δ*_C_ 39.5). ESIMS spectra were acquired on a Finnigan LCQ Deca mass spectrometer. HRESIMS spectra were measured with a UHR-QTOF maXis 4G (Bruker Daltonics) mass spectrometer.

### 4.2. Fungal Isolation, Identification, and Cultivation

The fungus *A. nomius* NC06 was isolated from the marine sponge *N. chaliniformis*. The sponge was collected in December 2015 from Mandeh Island, West Sumatra, Indonesia. The fungal strain was characterized by molecular identification of the 18S rRNA region. The sequence data were submitted to the GenBank with accession no MN242781. The fungal strain was deposited in the Laboratory of Biota Sumatra, Andalas University, Padang, Indonesia. Fermentation of the strain was carried out in 20 Erlenmeyer flasks, each containing 100 g of sterile rice media. The culture was kept under static conditions for 4–8 weeks or until the fungal mycelia fully grew on top of media.

### 4.3. Extraction and Isolation

Overgrown *A. nomius* NC06 on rice was extracted with 300 mL ethyl acetate (EtOAc). The EtOAc extract was evaporated in vacuo to obtain the crude extract (25.7 g). The crude extract was subjected to liquid–liquid partitioning between n-hexane and aqueous MeOH containing 10% of H_2_O. The MeOH extract (19.7 g) was subjected to vacuum liquid chromatography (VLC) on silica gel 60 by step gradient elution employing n-hexane, EtOAc, CH_2_Cl_2_, and MeOH to afford five fractions (F1 to F5). All fractions were screened for their cytotoxic activity against human colorectal adenocarcinoma (HT29) cells. Among the tested fractions, F3 eluted at EtOAc:DCM (1:1) showed the highest activity with an IC_50_ value of 2.59 µg/mL. F3 (3.95 g) was further chromatographed on silica gel 60 (VLC) by step gradient elution with n-hexane, EtOAc, CH_2_Cl_2_, and MeOH to yield 13 more subfractions (F3V1–F3V13). Compound **1** (7.23 mg) was obtained following the chromatographic purification of subfraction F3V7 on the Sephadex LH-20 column, which was eluted with CH_2_Cl_2_-MeOH (1:1 *v*/*v*), and final purification was achieved by semipreparative HPLC employing gradient elution with MeOH-H_2_O from 35% to 100% MeOH. Similarly, compound **2** (7.72 mg) and compound **3** (3.13 mg) were also purified using the same mobile phase. Furthermore, F4 was chromatographed on a silica column by gradient elution using CH_2_Cl_2_ and MeOH to afford 9 fractions. Subfraction F4V9 (112 mg) was subjected to semipreparative HPLC employing MeOH-H_2_O (from 35% to 100% MeOH) as a mobile phase to give compound **4** (52 mg). The structure of aspergillicin A was deduced by comparison of its spectral data from those found in the literature [31,32].

**Oxisterigmatocystin J (1).** Yellow amorphous solid; [*α*]20 °C—185 (*c* 0.05, CHCl_3_); UV (MeOH, PDA): λ_max_ 241.5 and 313.9 nm; ^1^H and ^13^C NMR data, see Table 1; HRESIMS *m*/*z* 355.0811 [M + H]^+^ (calc. for C_19_H_15_O_7_, 355.0812).

**Oxisterigmatocystin K** (**2**). Yellow amorphous solid; [*α*]20 °C—314 (*c* 0.05, CHCl_3_); UV (MeOH, PDA): λ_max_ 203.4, 237.5 and 314.3 nm; ^1^H and ^13^C NMR data, see Table 1; HRESIMS *m*/*z* 371.1127 [M + H]^+^ (calc. for C_20_H_19_O_7_, 371.1125).

**Oxisterigmatocystin L** (**3**). Yellow amorphous solid; [*α*]20 °C—181 (*c* 0.05, CHCl_3_); UV (MeOH, PDA): λ_max_ 202.9, 237.4 and 312.0 nm; ^1^H and ^13^C NMR data, see Table 1; HRESIMS *m*/*z* 371.1128 [M + H]^+^ (calc. for C_20_H_19_O_7_, 371.1125).

### 4.4. Mosher Ester Analysis of 1

Mosher’s method for compound **1** was adopted from a previously described procedure [28]. Two vials each containing (1.0 mg, 2.8 µmol) of compound **1** were each dissolved in 100 µL pyridine-d5. Each solution was transferred to an NMR tube. To the first solution, 10 µL (R)-MTPA-Cl (53.4 µmol) was added, while 10 µL (S)-MTPA-Cl (53.4 µmol) was added to the second solution. Both solutions were kept at room temperature for 3 h. Afterward, 500 µL of pyridine-d5 was added to each solution. ^1^H NMR spectra for both (S)- and (R)-MTPA ester derivatives were measured.

### 4.5. Cytotoxic Assay

All isolated compounds were tested against the HT29 colon cancer cell line by using the MTT method. This cell line was obtained from the Laboratory of Biotechnology and Cell Culture Pharmacy Faculty, International Islamic University Malaysia. HT29 was cultured by using DMEM Gibco^TM^ (high glucose). Cells were seeded in 96-well plates (density: 6 × 10^3^ cells/well) and incubated at 37 °C, 98% relative humidity, with 5% CO_2_, for 3 days. After cells were confluent in each well (70–80% confluent), compounds **1**–**4** were added to their respective wells with a concentration of 100, 10, 1, and 0.1 µg/mL. Then, 100 µL MTT (5 mg/mL) was added to each well and incubated for 4 h. Tecan Microplate was used for measuring the absorbance of the cells at 560 nm after treatment with all isolated compounds. DMSO was used for negative control and taxol was used for positive control.

### 4.6. Antibacterial Activity

Antibacterial activity was assessed by following the prototype of Balouiri et al. (2016) [32]. *E. coli* ATCC 25,922 and *S. aureus* ATCC 2592 were used as bacterial pathogens. Briefly, a paper disk (6 mm) was soaked with 10 μg/mL of all isolated compounds, **1**–**4**. Meanwhile, DMSO was used as negative control and 30 µg/disc chloramphenicol was used as a positive control. The diameter of the zone (mm) of inhibition was measured after incubation for 24 h.

### 4.7. Apoptotic Cells Using AO–PI Double Staining

Acridine orange (AO) and propidium iodide (PI) were used for quantifying viable, apoptotic, and necrotic cells of HT 29. The double staining method was a standard procedure and examined under a fluorescence microscope. As many as 1 × 10^5^ cells in 4 mL were seeded in a T25 cm^3^ flask. After 24 h of incubation, the medium in each well was removed and replaced with the IC_50_ concentrations of compounds **1**–**2** and **4** (6.28, 15.14, 1.63 µg/mL) as treated cells, and untreated cells were cells without any treatment. Once the cytotoxic compound was added to the medium, the cells were incubated at 37 °C, 98% relative humidity, with 5% CO_2_, for 24 h. After 24 h incubation, the cells were washed with 1 mL PBS. The cells were then trypsinized and centrifuged at 1000× *g* rpm for 5 min. The AO–PI double stain was dissolved with PBS and then added to the cell pellet for the staining step. The suspension (50 µL) of stained cells was dropped onto a glass slide and covered with a coverslip. The observation was conducted with a fluorescence microscope at 400 magnification within 30 min before the fluorescence faded. Viable, apoptotic, and necrotic cells were quantified in a population of 200 cells. The results were expressed as a proportion of the total number of the cells examined [26].

## Figures and Tables

**Figure 1 marinedrugs-19-00631-f001:**
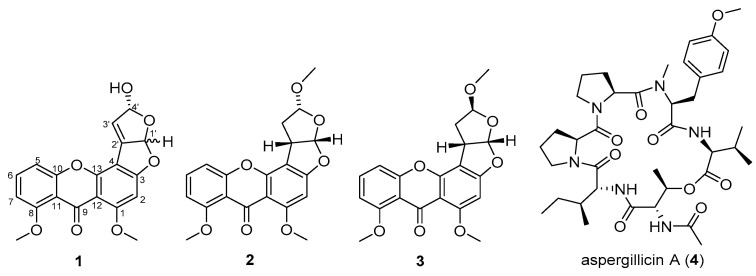
Structures of isolated compounds **1–4.** Stereochemistry shown for compounds **2** and **3** is relative.

**Figure 2 marinedrugs-19-00631-f002:**
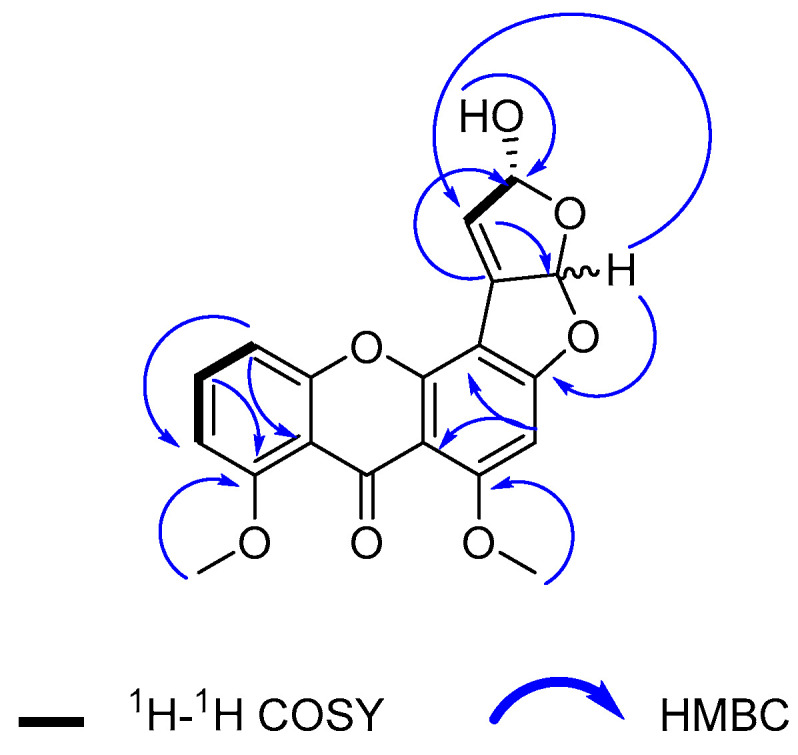
Key ^1^H-^1^H COSY and HMBC (H to C) correlations observed for compound **1**.

**Figure 3 marinedrugs-19-00631-f003:**
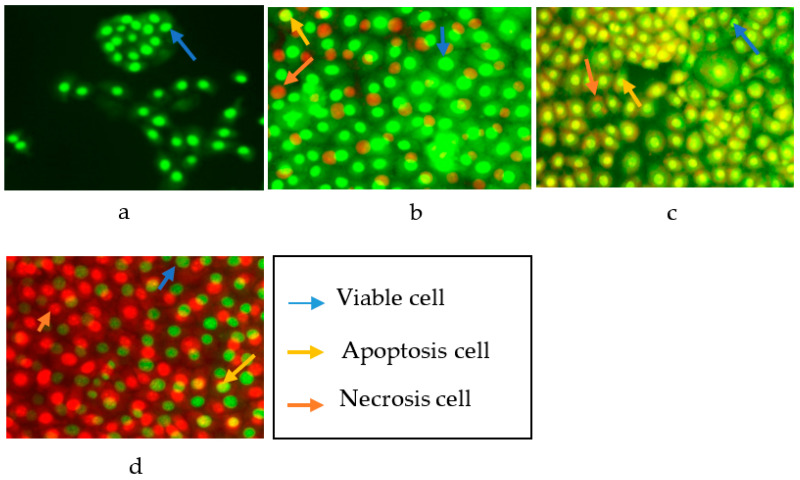
Cells after staining with AO–PI: (**a**) untreated HT 29 cells; (**b**) HT 29 cells treated with compound 1; (**c**) HT 29 cells treated with compound 2; (**d**) HT 29 cells treated with compound 4.

**Table 1 marinedrugs-19-00631-t001:** ^1^H (600 MHz) and ^13^C (150 MHz) NMR data for compounds **1–3 in DMSO-*d*_6_**.

Position	1		2		3	
	*δ*_C_, Type	*δ*_H_ (*J* in Hz)	*δ*_C_, Type	*δ*_H_ (*J* in Hz)	*δ*_C_, Type	*δ*_H_ (*J* in Hz)
1	162.5, C		161.9, C		162.1, C	
2	91.0, CH	6.62, s	90.3, CH	6.50, s	90.3, CH	6.55, s
3	163.5, C		163.0, C		162.9, C	
4	108.3, C		107.0, C		107.9, C	
5	106.8, CH	6.94, dd (8.5, 0.9)	108.9, CH	7.02, dd (8.4, 0.9)	108.7, CH	7.04, d (8.3)
6	134.3, CH	7.65, t (8.4)	134.1, CH	7.61, t (8.3)	133.9, CH	7.62, t (8.4)
7	109.1, CH	7.08, dd (8.4, 0.9)	106.8, CH	6.93, dd (8.4, 0.9)	106.6, CH	6.93, d (8.3)
8	155.9, C		159.6, C		159.6, C	
9	173.0, C		173.2, C		173.1, C	
10	159.6, C		156.0, C		155.8, C	
11	113.9, C		112.9, C		113.1, C	
12	108.2, C		107.5, C		106.0, C	
13	152.7, C		152.2, C		152.7, C	
1′	117.3, CH	6.41, s	113.3, CH	6.56, d (6.1)	111.0, CH	6.48, d (5.9)
2′	89.2, C		42.0, CH	4.19, dd (9.3, 6.2)	41.8, CH	4.24, ddd (9.2, 5.9, 3.4)
3′	148.3, CH	6.77, d (2.8)	36.7, CH_2_	2.39, ddd (13.4, 9.5, 5.2) 2.23, d (13.3)	36.0, CH_2_	2.35, dd (13.2, 4.8, 3.4) 2.25, ddd (13.5, 9.0, 5.3)
4′	107.0, CH	5.71, d (2.8)	106.3, CH	5.25, d (5.0)	106.1, CH	5.17, t (5.0)
1-OMe	56.5, CH_3_	3.84, s	56.3, CH_3_	3.81, s	56.1, CH_3_	3.82, s
8-OMe	56.1, CH_3_	3.85, s	56.1, CH_3_	3.85, s	55.8, CH_3_	3.85, s
4′-OH		6.50, s				
4′-OMe			54.5, CH_3_	3.09, s	55.7, CH_3_	3.36, s

**Table 2 marinedrugs-19-00631-t002:** Cytotoxic activity of all compounds against HT29 colon cancer cells.

Compound No	Compound Name	IC50 (µM)
**1**	oxisterigmatocystin J	6.28
**2**	oxisterigmatocystin K	15.14
**3**	oxisterigmatocystin L	988.05
**4**	aspergillicin A	1.63
	Taxol	0.48

**Table 3 marinedrugs-19-00631-t003:** Percentages of viable, apoptotic, and necrotic cells after treatment for 24 h.

Cytotoxic Compound	Viable Cell (%)	Apoptotic Cell (%)	Necrotic Cell (%)
**1**	64.53 ± 3.05	30.65 ± 3.22	4.81 ± 1.02
**2**	36.49 ± 4.08	59.38 ± 4.88	4.12 ± 1.37
**4**	25.24 ± 4.21	4.80 ± 0.98	69.95 ± 4.16

Data are shown as mean ± standard error (*n* = 3).

## Data Availability

Data are contained within the article or Appendix A.

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
