# Peer review of "Apoptotic Activity of New Oxisterigmatocystin Derivatives from the Marine-Derived Fungus Aspergillus nomius NC06"

_marinedrugs, 2021, doi:10.3390/md19110631_

Round 1

Reviewer 1 Report

Handayani, Edrada-Ebel and colleagues describe three previously unreported sterigmatocystin derivatives obtained from a MeOH extract of a marine-derived strain of Aspergillus nomius, as well their cytotoxicity towards HT 29 cells and pro-apoptotic effects.

While the novel sterigmatocystin derivatives (oxisterigmatocystins J-L) are ‘known unknowns’ i.e., bearing minor structural variations/substituents, the content of the manuscript fits within the scope of Marine Drugs. Nevertheless, as below detailed, authors should consider a series of issues.

First, several typos and syntax errors should be revised as (for example) in:

Title; Abstract, Keywords: Revise “Aspergilllus” to “Aspergillus”.

Lines 22-23:…spectroscopy as well as by high-resolution mass spectrometry.”.

Lines 28-29: Revise “…were able to induce the apoptotic cell death in HT 29 cells”.

Lines 43-45: “The fungal genus of Aspergillus has been repeatedly described in the literature for the potent cytotoxic activity of its extracts and metabolites. One of the compounds isolated from this genus is an oxisterigmatocystin derivative [16,17].”

The sentences are unclear!

Line 51: Consider revising to “…from the fungus Aspergillus sp. F40 derived from samples of a Callyspongia sp. sponge”.

Line 52: Revise “oxisterigmatocysin” to “oxisterigmatocystin”.

Line 152: Revise “camphonate” to “camphonata”.

Further examples can be found throughout the whole entire manuscript, authors being advised to have it edited by an English native speaker or editing service.

Structures of the novel constituents agree with the 1D and 2D NMR data, but the configurations at C4’ are wrongly assigned in the text; revise β- to α-configuration (Compound 2; Line 88) and α- to β-configuration (Compound 3; Line 91).

In order to enable the comparison with reference drugs, it would be more convenient to present the IC50 values (Table 2) estimated for the cytotoxicity upon HT29 cells in units of molar concentration. Furthermore, as taxol has been used as a ‘positive control’ (rather as a reference drug) (Line 243), authors should present the IC50 value, allowing to know the true potential of the isolated constituents.

Discussion dealing with the cytotoxic effects of structurally unrelated compounds (Lines 148-155) is useless as it does not allow any comparison or to deliver outcomes on structural requirements favouring the cytotoxicity of upon HT 29 cells. In this matter, authors are advised to deliver a brief discussion on the cytotoxicity of structurally related compounds (i.e., bearing both the xanthone nucleus and a bifuran structure) upon HT 29 cells.    

Finally, a conclusion remains to be delivered.

Reviewer 2 Report

The article concerns the study of production of secondary metabolites by the fungi Aspergilllus nomius NC06 collected from marine sponge Neopetrosia chaliniformis. The new phenolic compounds, namely oxisterigmatocystin J, K, and L and known aspergillicin A have been isolated. The structures were elucidated or identified using 1D and 2D NMR spectroscopy and HRMS. The isolated compounds were tested for cytotoxic activity against HT 29 colon cancer cells using MTT analysis. Three active compounds were studied on apoptosis induction at ID50 cytotoxic concentrations using fluorescence microscopy by a double staining (acridine orange + propidium iodide) method. Two active new compounds were good inductors of apoptotsis of HT 29 cells.

The biological part is good and well written however the phrase “induce the apoptotic” (Line 29) should be replaced with “induce the apoptosis”. However, the chemical part is not satisfactory. The structural elucidation is described too briefly and unclear. It is not understood what is the source of the molecular formulae of the isolated compounds, what is the certain ion peak, its certain m/z value (experimental and calculated). The author listed the structural fragments identified by 1D NMR of the first substance without values of corresponding chemical shifts, sort of spectra etc. I also don’t understand what is “at ca. 13 ppm.”

There is no evidence of the relative and absolute configurations of stereocenters at C-1’ in all the new compounds. The authors should provide additional evidences such as X-ray data, additional 2D NMR experiments etc. or just depict these centers as undetermined.

The chemical part should be rewritten as a whole along with the published similar articles as a sample.

My recommendation is: major revision.

Reviewer 3 Report

This manuscript reported the chemical and biological investigation of four secondary metabolites from sponge-derived fungus Aspergillus nomius. Although the isolated compounds exhibited cytotoxic activities against HT29, the structures were less of novelty. Several structural similar analogues exhibited a more significant cytotoxic activity. Moreover, there are many issues in the structural identification which need major revision or further experimental data to support. Thus, I do not think the manuscript is suitable for publication in its current form and more supporting data and a major revision are needed:

  1. Page 2, line 73: the Mosher's method is confused. Line 78: what does the "ΔδR-S" mean? Is it the R-/S-MTPA esters? If so, the results are incorrect. The compounds reacted with R-/S-MTPA-Cl will afford S-/R-MTPA ester, respectively, and the the ΔδS-R of the ester should be used to identified the configuration. In section 3.4, the authors described that the MTPA-Cl was used as the Mosher reagents but in line 75, the MTPA was mentioned as Mosher reagent.
  2. Page 2, line 80: how did the author assign the R configuration of C-1 without any evidence?
  3. Page 2, line 86: the authors established the absolute configuration of compound 2 and 3 by comparing their NMR data to those of the known oxisterigmatocystin C and B, respectively. However, the similar chemical shifts and coupling constants could only lead to determined the relative configuraion. The absolute configuration should be established by more spectroscopic data comparison such as ECD and optical rotation.
  4. Page 3, line 98: the HMBC correlation in Figure 2 is disordered, which should be re-drawn carefully.
  5. All the compound number should be bold. The reference format should be uniform and all latin names should be intalic.
  6. In the supporting information, the HMBC spectra are not complete, the author should provide a new one.

Round 2

Reviewer 2 Report

Because the authors corrected the errors noted in my report I recommend to publish the article.

Author Response

Thanks so much for your reports and comments in making this manuscript better.

Reviewer 3 Report

Recommendation:

Comments:
This manuscript has been revised partly according to my suggestions from the reviewer. The confused writing and figures in the main text have been improve. However, there are still some mistakes need to be revised:

  1. The HMBC spectra in the supporting information have not been revised. Please provide the correct figures.
  2. The author described that the through-space distance between H-1’ and H-4’ are 3.814 and 3.448Å, respectively. The evidence or the analysis process should be provided.
  3. The coupling constants between the coupled protons should be revised to the same value in both table 1 and the main text.
  4. In table 3, a blank should be inserted between numbers and symbols.
  5. According to the compound data in section 4.3, the optical rotation values and the quality of compounds 2 and 3 indicated that the ECD spectra could be collected. Why did the author said that the it is not possible to measure the ECD.

Author Response

  1. The HMBC spectra in the supporting information have not been revised. Please provide the correct figures.

 All HMBC and HSQC spectra were revised as recommended. The 1D (1H and 13C) spectra were added on the respective X and Y axis of the HMBC and HSQC spectra. Our apologies, we did not understand what the reviewer was asking as all 2D heteronuclear spectra were shown between (HSQC: 0-150 ppm for 13C and 0-11ppm for 1H; HMBC: 20-180ppm for 13C and 0-9ppm for 1H)

  1. The author described that the through-space distance between H-1’ and H-4’ are 3.814 and 3.448Å, respectively. The evidence or the analysis process should be provided.

The following were added on lines 83 to 89.

The observed nOe was inconclusive because in both cases whether H-1’ has the R or S configuration, the through-space distance between H-1’ and H-4’ are 3.814 and 3.448Å, respectively, which are both less than the internuclear separation of 5Å that is the minimum requirement to identify spatially close pairs of nuclei. These through-space distances were determined by MM2 calculation at minimum energy of 84.5646 and 81.4504 kcal/mol, respectively using PerkinElmer’s 3D ChemDraw v 18.2.

  1. The coupling constants between the coupled protons should be revised to the same value in both table 1 and the main text.

The value of coupling constants in the text were rounded off to its integers.  To avoid confusion the sentences were revised:

Lines 97-100: While again a similar ABC system was observed for H-5, H-6, and H-7 in both congeners 2 and 3 at δH 7.02/7.04 (d, J = 8.4/8.3 Hz), 7.61/7.62 (t, J = 8.3/8.4 Hz), 6.93 (d, J = 8.4/8.3 Hz), respectively with differences in chemical shifts between ±0.01 and 0.02 ppm between the two isolated derivatives.

Lines 109-110: The coupling constant between H-1’ and H-2’ of compounds 2 and 3 was at 6 Hz (with actual values at 6.1 and 5.9 Hz, respectively) that implied the cis configuration for the two vicinal protons and indicated a 1’S, 2’S stereochemistry [18].

Lines 119-121: The coupling constant between H-2’ at 4.19 ppm and H-3’A at 2.39 ppm gave 9 Hz (with actual values at 9.3 and 9.5 Hz, respectively) for a dihedral angle of about 20o [24].

All other coupling constants in the text were double checked to be identical as presented in Table 1, which were highlighted in blue.

  1. In table 3, a blank should be inserted between numbers and symbols.

We have inserted a blank between numbers and symbols in table 3, which is marked with a blue highlight.

Table 3. Percentages of viable, apoptotic, and necrotic cells after treatment 24 h.

Cytotoxic compound

Viable cell (%)

Apoptotic cell (%)

Necrotic cell (%)

1

64.53 ± 3.05

30.65 ± 3.22

4.81 ± 1.02

2

36.49 ± 4.08

59.38 ± 4.88

4.12 ± 1.37

4

25.24 ± 4.21

4.80 ± 0.98

69.95 ± 4.16

Data are shown as mean ± standard error (n=3).

  1. According to the compound data in section 4.3, the optical rotation values and the quality of compounds 2 and 3 indicated that the ECD spectra could be collected. Why did the author said that the it is not possible to measure the ECD.

To date, the compounds have already been used up for the bioassays. The authors do not have any compounds left to measure the ECD. The referee is right, we should have measured the ECD in the first place and  this should have been possible. BUT ECD measurement should have been done prior to doing any bioassay the authors have performed but now is too late, we do not have any left of the compounds to measure the ECD. We also do not have any funding anymore to recollect, reisolate, and purify again so the authors can only show the relative configuration in this paper.

Round 3

Reviewer 3 Report

The manuscript has been modified as much as possible according to my suggestion, which is now suitable for publication.

This manuscript is a resubmission of an earlier submission. The following is a list of the peer review reports and author responses from that submission.